# Depth Induced Regression Medians and Uniqueness

**Yijun Zuo** 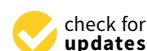

Department of Statistics and Probability, Michigan State University, East Lansing, MI 48824, USA; zuo@msu.edu

**Abstract:** The notion of median in one dimension is a foundational element in nonparametric statistics. It has been extended to multi-dimensional cases both in location and in regression via notions of data depth. Regression depth (RD) and projection regression depth (PRD) represent the two most promising notions in regression. Carrizosa depth $D_C$ is another depth notion in regression. Depth-induced regression medians (maximum depth estimators) serve as robust alternatives to the classical least squares estimator. The uniqueness of regression medians is indispensable in the discussion of their properties and the asymptotics (consistency and limiting distribution) of sample regression medians. Are the regression medians induced from RD, PRD, and $D_C$ unique? Answering this question is the main goal of this article. It is found that only the regression median induced from PRD possesses the desired uniqueness property. The conventional remedy measure for non-uniqueness, taking average of all medians, might yield an estimator that no longer possesses the maximum depth in both RD and $D_C$ cases. These and other findings indicate that the PRD and its induced median are highly favorable among their leading competitors.

**Keywords:** uniqueness; regression depth; maximum depth estimator; regression median; robustness

## 1. Introduction

Regular univariate sample median defined as the innermost (deepest) point of a data set is unique (If the sample median is defined to be the point $\theta$ that minimizes the sum of its distances to sample points (i.e., $\theta = \arg\min_{\theta \in \mathbb{R}^1} \sum_{i=1}^{n} |\theta - x_i|$, where $x_i, i = 1, \cdots, n$ are the given $n$ sample points in $\mathbb{R}^1$), then it is not unique. However, to overcome this drawback, conventionally it is defined as $\theta = \text{Median}\{x_i\} := x_{(\lfloor \frac{n+1}{2} \rfloor)} + x_{(\lfloor \frac{n+2}{2} \rfloor)}/2$, where $x_{(1)} \leq x_{(2)} \leq \cdots \leq x_{(n)}$ are ordered values of $x_i$'s and $\lfloor \cdot \rfloor$ is the floor function. Namely, it is the innermost point (from both left and right direction) or the average of two deepest sample points. Hence, it is unique). The population median defined as the $\frac{1}{2}$-th quantile (Recall, for any univariate distribution function $F$, and for $0 < p < 1$, the quantity $F^{-1}(p) := \inf\{x : F(x) \geq p\}$ is called the $p$th *quantile* or *fractile* of $F$ (see page 3 of Serfling (1980) [1])) of the underlying distribution (there are other versions of definition) is also unique. The most outstanding feature of the univariate median is its robustness. In fact, among all translation equivariant location estimators, it has the best possible breakdown point (Donoho (1982) [2]) (and the minimum maximum bias if underlying distribution has a unimodal symmetric density (Huber (1964) [3]). Besides serving as a promising robust location estimator, the univariate median also provides a base for a center-outward ordering (in terms of the deviations from the median), an alternative to the traditional left-to-right ordering.

To extend the univariate median to multidimensional settings and to share its outstanding robustness property and an alternative ordering scheme is desirable for multidimensional data. One approach, among others, is via notions of data depth. General notions of data depth have been increasingly pursued and studied (Liu, et al. (1999) [4], Zuo and Serfling (2000) (ZS00) [5]) since the pioneer proposal of Tukey (1975) [6] (see Donoho and Gasko (1992) [7]). Besides Tukey depth, another prevailing depth, among others, is the projection depth (PD) [5] (Liu (1992) [8], and Zuo (2003) [9]).

Depth notions in location have also been extended to regression. Regression depth (RD) of Rousseeuw and Hubert (1999) (RH99) [10], the most famous, exemplifies a direct extension of Tukey location depth to regression. Projection regression depth (PRD) of Zuo (2018a) (Z18a) [11] is another example of the extension of prominent PD in location to regression. The RD and PRD represent the two leading notions of depth in regression ([11]) which satisfy desirable axiomatic properties. Carrizosa depth $D_C$ (Carrizosa (1996) (C96)) [12] (defined in Section 2.2) is one of the other notions of depth in regression ([11]). One of the outstanding advantages of depth notions is that they can be directly employed to introduce median-type deepest estimating functionals (or estimators in the empirical case) for the location or regression parameters in a multi-dimensional setting based on a general min-max stratagem. The maximum (deepest) regression depth estimator (also called regression median) serves as a *robust* alternative to the classical least squares or least absolute deviations estimator for the unknown parameters in a general linear regression model:

$$y_i \;\; = \;\; \mathbf{x}_i^\top \boldsymbol{\beta} + e_i, \text{ for } i = 1, \cdots, n, \tag{1}$$

where $\top$ denotes the transpose of a vector, and vector $\mathbf{x}_i = (1, x_{i1}, \cdots, x_{i(p-1)})^\top$ and parameter vector $\boldsymbol{\beta} = (\beta_1, \cdots, \beta_p)$ are in $\mathbb{R}^p$ ($p \geq 2$), and $e_i$ is a random variable in $\mathbb{R}$. One can regard the observations $(y_i, \mathbf{x}_i^\top)$ as a sample from random vector $(y, \mathbf{x}^\top) \in \mathbb{R}^{p+1}$.

Robustness of the median induced from RD and PRD have been investigated in Van Aelst and Rousseeuw (2000) (VAR00) [13] and Zuo (2018b) [14], respectively. These medians, just like their location or univariate counterpart, indeed possess high breakdown point robustness.

Regression median, as the deepest regression hyperplane, just like their location or univariate counterpart, is expected to be unique because non-uniqueness would result in vagueness in the inference (prediction and estimation) via regression median. Uniqueness is the indispensable feature and axiomatic property when one (i) investigates the population median, or (ii) deals with the convergence in probability or in distribution of the sample regression median to its inevitably unique population version (iii) it is also an essential property in the computation of the sample regression medians for the convergence of approximate algorithms. The uniqueness issue of multidimensional location medians has been addressed in Zuo (2013) [15].

Are the medians induced from regression depth notions via the min-max scheme generally unique? Answering this question is the goal of this article. It turns out that the regression depth-induced medians are not necessarily unique. The conventional remedy measure for this issue is taking average of all. It, however, might not work (in the sense that the resulting estimator might no longer possess the maximum depth) for both RD of RH99 and $D_C$ of C96. On the other hand, PRD-induced regression medians are unique.

The rest of article is organized as follows. Section 2 introduces leading regression depth notions and induced medians and show these medians indeed recover the regular univariate sample median in the special univariate case. Empirical examples of regression depth and medians and their behavior are illustrated in Section 3. Section 4 establishes general results on uniqueness of regression medians. Brief concluding remarks in Section 5 end the article.

## 2. Maximum Depth Functionals (Regression Medians)

Let $D(\boldsymbol{\beta}; P)$ be a generic non-negative functional on $\mathbb{R}^p \times \mathcal{P}$, where $\boldsymbol{\beta} \in \mathbb{R}^p$ and $\mathcal{P}$ is a collection of distributions $F_Z$ of $Z = (y, \mathbf{x}^\top)^\top \in \mathbb{R}^{p+1}$ ($F_Z$ and P are used interchangeably).

If $D(\boldsymbol{\beta}; P)$ satisfies four axiomatic properties: **(P1)** (regression, scale and affine) invariance; **(P2)** maximality at center; **(P3)** monotonicity relative to any deepest point and **(P4)** vanishing at infinity, then it is called a regression depth functional (see [11] for details). The maximum regression depth functional, or the regression median, can be defined as

$$\boldsymbol{\beta}^*(F_Z) := \arg\max_{\boldsymbol{\beta} \in \mathbb{R}^p} D(\boldsymbol{\beta}; F_Z). \tag{2}$$

Note that $\boldsymbol{\beta}^*$ might not be unique, and a conventional remedy measure is to take the average of all maximum depth points. Unfortunately, this could lead to a scenario where the resulting functional (or estimator) might not have the maximum depth any more. For detailed discussions on $D(\boldsymbol{\beta}; F_Z)$ and $\boldsymbol{\beta}^*(F_Z)$, see [11]. In the following we elaborate three examples.

### 2.1. Median Induced from Regression Depth of RH99

**Definition 1.** *For any $\boldsymbol{\beta} \in \mathbb{R}^p$ and the joint distribution $P$ of $(y, \mathbf{x}^\top)$ in (1), [10] defined the regression depth of $\boldsymbol{\beta}$, denoted hence by $RD_{RH}(\boldsymbol{\beta}; P)$, to be the minimum probability mass that needs to be passed when tilting (the hyperplane induced from) $\boldsymbol{\beta}$ in any way until it is vertical. The maximum regression depth functional $\boldsymbol{\beta}^*_{RD_{RH}}$ (regression median) is defined as*

$$\boldsymbol{\beta}^*_{RD_{RH}}(P) = \underset{\boldsymbol{\beta} \in \mathbb{R}^p}{\operatorname{argmax}} \, RD_{RH}(\boldsymbol{\beta}; P) \tag{3}$$

The $RD_{HR}(\boldsymbol{\beta}; P)$ definition above is rather abstract and not easy to comprehension, many characterizations of it, or equivalent definitions, have been given in the literature though, see, e.g., [11] and references cited therein.

### 2.2. Median Induced from Carrizosa Depth of C96

Among regression depth notions investigated in [11], Carrizosa depth $D_C(\boldsymbol{\beta}; P)$

$$D_C(\boldsymbol{\beta}; P) = \inf_{\boldsymbol{\alpha} \in \mathbb{R}^p} P(|r(\boldsymbol{\beta})| \leq |r(\boldsymbol{\alpha})|), \tag{4}$$

for any $\boldsymbol{\beta} \in \mathbb{R}^p$ and underlying probability measure $P$ associated with $(y, \mathbf{x}^\top)$, was a pioneer regression depth notion introduced in [12] and thoroughly investigated in [11], where $r(\boldsymbol{\gamma}) = y - \mathbf{x}^\top \boldsymbol{\gamma}$. As characterized in [11] (see Proposition 2.2 there), it turns out that $(p \geq 2)$

$$D_C(\boldsymbol{\beta}; P) = P(r(\boldsymbol{\beta}) = 0). \tag{5}$$

The maximum regression depth functional (or regression median) was then defined as

$$\boldsymbol{\beta}^*_{D_C}(P) = \arg\max_{\boldsymbol{\beta} \in \mathbb{R}^p} D_C(\boldsymbol{\beta}; P). \tag{6}$$

As shown in [11], $\boldsymbol{\beta}^*_{D_C}$ always exists if the assumption: **(A)** $P(H_v) = 0$, for any vertical hyperplane $H_v$, holds. Unfortunately, as $D_C$ violates **(P3)** generally (see [11]), we will not focus on it in the sequel. On the other hand, under **(A)** $RD_{RH}$ above satisfies **(P1)–(P4)**.

### 2.3. Median Induced from Projection Regression Depth of Z18a

Hereafter, assume that $R$ is a univariate regression estimating functional which satisfies

**(A1)** regression, scale and affine equivariant. That is, respectively,

$R(F_{(y+xb, \, x)}) = R(F_{(y, \, x)}) + b, \; \forall \, b \in \mathbb{R}^1$, and

$R(F_{(sy, \, x)}) = sR(F_{(y, \, x)}), \; \forall \, s \in \mathbb{R}^1$, and

$R(F_{(y, \, ax)}) = a^{-1}R(F_{(y, \, x)}), \; \forall \, a(\neq 0) \in \mathbb{R}^1$.

where $x, y \in \mathbb{R}^1$ are random variables. Throughout, the lower case $x$ stands for a variable in $\mathbb{R}^1$ while the bold $\mathbf{x}$ for a vector in $\mathbb{R}^p$ $(p > 1)$. $F_{(x_1, x_2)}$ is the distribution of vector $(x_1, x_2)$.

**(A2)** $\sup_{\mathbf{v} \in \mathbb{S}^{p-1}} |R(F_{(y, \, \mathbf{x}^\top \mathbf{v})})| < \infty$, where $\mathbb{S}^{p-1} := \{\mathbf{u} \in \mathbb{R}^p, \|\mathbf{u}\| = 1\}$.

**(A3)** $R(F_{(y-\mathbf{x}^\top \boldsymbol{\beta}, \, \mathbf{x}^\top \mathbf{v})})$ is continuous in $\boldsymbol{\beta}$ and $\mathbf{v}$, and quasi-convex in $\boldsymbol{\beta}$, for $\boldsymbol{\beta} \in \mathbb{R}^p$, $\mathbf{v} \in \mathbb{S}^{p-1}$.

Let $S$ be a positive scale estimating functional that is *scale equivariant* and *location invariant*.

$R$ will be restricted to the form $R(F_{(y-\mathbf{x}\top\boldsymbol{\beta},\,\mathbf{x}\top\mathbf{v})}) = T(F_{(y-\mathbf{x}\top\boldsymbol{\beta})/(\mathbf{x}\top\mathbf{v})})$ and $T$ will be a univariate location functional that is location, scale and affine equivariant (see pages 158–159 of Rousseeuw and Leroy (1987) (RL87) [16] for definitions). Hereafter we assume that **(A0)** $P(\mathbf{x}\top\mathbf{v} = 0) = 0$ for any $\mathbf{v} \in \mathbb{S}^{p-1}$ (see (I) of Remarks 4.1 for the explanations).

Examples of $T$ include, among others, the mean, weighted mean, and quantile functionals. A particular example of $R(F_{(y-\mathbf{x}\top\boldsymbol{\beta},\,\mathbf{x}\top\mathbf{v})})$ is $\mathrm{Med}_{\mathbf{x}\top\mathbf{v}\neq 0}\left(F_{(y-\mathbf{x}\top\boldsymbol{\beta})/\mathbf{x}\top\mathbf{v}}\right)$, where Med stands for the median functional. Typical examples of S include the standard deviation and weighted deviation functionals (Wu and Zuo (2008) [17]) and the median of absolute deviations (MAD) functional.

Equipped with a pair of $T$ and $S$, we can introduce a corresponding projection based regression estimating functional. By modifying a functional in Marrona and Yohai (1993) [18] to achieve scale equivarance, [11] defined

$$\mathrm{UF}_{\mathbf{v}}(\boldsymbol{\beta};\,F_{(y,\,\mathbf{x}^\top)},T) := |T(F_{(y-\mathbf{x}\top\boldsymbol{\beta})/\mathbf{x}\top\mathbf{v}})|/S(F_y), \tag{7}$$

which represents unfitness of $\boldsymbol{\beta}$ at $F_{(y,\,\mathbf{x}^\top)}$ w.r.t. $T$ along the $\mathbf{v} \in \mathbb{S}^{p-1}$. If $R$ is a *Fisher consistent* regression estimating functional, then $T(F_{(y-\mathbf{x}\top\boldsymbol{\beta}_0)/\mathbf{x}\top\mathbf{v}}) = 0$ for some $\boldsymbol{\beta}_0$ (the true parameter of the model) and $\forall\,\mathbf{v} \in \mathbb{S}^{p-1}$. Thus overall, one expects $|T|$ to be small and close to zero for a candidate $\boldsymbol{\beta}$, independent of the choice of $\mathbf{v}$ and $\mathbf{x}\top\mathbf{v}$. The magnitude of $|T|$ measures the unfitness of $\boldsymbol{\beta}$ along the $\mathbf{v}$. Taking the supremum over all $\mathbf{v} \in \mathbb{S}^{p-1}$ yields

$$\mathrm{UF}(\boldsymbol{\beta};\,F_{(y,\,\mathbf{x}^\top)},T) = \sup_{\|\mathbf{v}\|=1} \mathrm{UF}_{\mathbf{v}}(\boldsymbol{\beta};\,F_{(y,\,\mathbf{x}^\top)},T), \tag{8}$$

the *unfitness* of $\boldsymbol{\beta}$ at $F_{(y,\,\mathbf{x}^\top)}$ w.r.t. $T$. Now applying the min-max scheme, [11] obtained the *projection regression estimating functional* (also denoted by $\boldsymbol{\beta}^*_{\mathrm{PRD}}$) w.r.t. the pair $(T,S)$

$$\boldsymbol{\beta}^*(F_{(y,\,\mathbf{x}^\top)},T) = \arg\min_{\boldsymbol{\beta}\in\mathbb{R}^p} \mathrm{UF}(\boldsymbol{\beta};\,F_{(y,\,\mathbf{x}^\top)},T) \tag{9}$$

$$= \mathrm{argmax}_{\boldsymbol{\beta}\in\mathbb{R}^p} \mathrm{PRD}\left(\boldsymbol{\beta};\,F_{(y,\,\mathbf{x}^\top)},T\right),$$

where the *projection regression depth* (PRD) functional was defined in [19] as

$$\mathrm{PRD}\left(\boldsymbol{\beta};\,F_{(y,\,\mathbf{x}^\top)},T\right) = \left(1 + \mathrm{UF}(\boldsymbol{\beta};\,F_{(y,\,\mathbf{x}^\top)},T)\right)^{-1}, \tag{10}$$

Just like $S$ (which is for achieving scale invariance and is nominal), $T$ sometimes is also suppressed in above functionals for simplicity. The authors of [11] showed that PRD satisfies **(P1)–(P4)**.

For robustness consideration, in the sequel, $(T,S)$ is the fixed pair $(\mathrm{Med},\mathrm{MAD})$, unless otherwise stated. Hereafter, we write $\mathrm{Med}(Z)$ rather than $\mathrm{Med}(F_Z)$. For this special choice of $T$ and $S$, we have that

$$T(F_{(y-\mathbf{x}\top\boldsymbol{\beta})/\mathbf{x}\top\mathbf{v}}) = \mathrm{Med}_{\mathbf{x}\top\mathbf{v}\neq 0}\left(\frac{y-\mathbf{x}\top\boldsymbol{\beta}}{\mathbf{x}\top\mathbf{v}}\right),$$

$$S(F_y) = \mathrm{MAD}(F_y).$$

To end this section, we show that the three maximum depth estimators above indeed deserve to be called regression median since they recover the regular univariate sample median in the special univariate case. (The result below also holds true for the population case).

**Proposition 1.** *For univariate data, the $\boldsymbol{\beta}^*_{RD_{RH}}$, $\boldsymbol{\beta}^*_{DC}$ and $\boldsymbol{\beta}^*_{PRD}$ all recover the univariate sample median.*

**Proof.** (i) For $\boldsymbol{\beta}^*_{RD_{RH}}$, this has already been discussed and claimed in [10] (page 390). So we only need to focus on the other two.

(ii) For $\boldsymbol{\beta}^*_{D_C}$, we no longer can use (5) and have to invoke (4). Note that $r(\beta) = y - \beta$ in this case (no slope term any more).

$$
\begin{aligned}
D_C(\beta, P) &= \inf_{\alpha \in \mathbb{R}^1} P(|y - \beta| \le |y - \alpha|) \\
&= \min \Big\{ \inf_{\alpha > \beta} P(|y - \beta| \le |y - \alpha|), \ \inf_{\alpha \le \beta} P(|y - \beta| \le |y - \alpha|) \Big\} \\
&= \min \big\{ P(y \le \beta), \ P(y \ge \beta) \big\}.
\end{aligned}
$$

That is, $D_C(\beta, P_n) = \min \big\{ \sum_{i=1}^n I(y_i \le \beta)/n, \ \sum_{i=1}^n I(y_i \ge \beta)/n \big\}$. The latter immediately leads to $\beta^*_{D_C} = \mathrm{Med}_i\{y_i\}$ (the average of all solutions).

(iii) For $\boldsymbol{\beta}^*_{PRD}$, first we note that (without loss of generality, assume that $S(F_y) = 1$)

$$
\boldsymbol{\beta}^*_{PRD} = \arg \min_{\boldsymbol{\beta} \in \mathbb{R}^p} \sup_{\mathbf{v} \in \mathbb{S}^{p-1}} \Big| \mathrm{Med}_i\{ \frac{y_i - \mathbf{x}^\top_i \boldsymbol{\beta}}{\mathbf{x}^\top_i \mathbf{v}} \} \Big|.
\tag{11}
$$

When $p = 1$, it reduces to the following

$$
\beta^*_{PRD} = \arg \min_{\beta \in \mathbb{R}} \sup_{v = \pm 1} \Big| \mathrm{Med}_i\{ \frac{y_i - \beta}{v} \} \Big|.
\tag{12}
$$

It is readily seen that

$$
\beta^*_{PRD} = \arg \min_{\beta \in \mathbb{R}} \big| \mathrm{Med}_i\{y_i - \beta\} \big| = \arg \min_{\beta \in \mathbb{R}} \big| \mathrm{Med}_i\{y_i\} - \beta \big|,
\tag{13}
$$

where the first equality follows from (12) and the oddness of median operator, the second one follows from the translation equivalence (see page 249 of [16] for definition) of the median as a location estimator. The last display means that $\beta^*_{PRD}$ recovers the sample median. $\square$

## 3. Examples of Regression Depths and Regression Medians

For a better comprehension of depth notions and depth-induced medians in the last section, we present empirical examples below. We will confine attention to RD and PRD only since $D_C(\boldsymbol{\beta}, P)$ is just the probability mass carried by the hyperplane determined by $y = \mathbf{x}^\top \boldsymbol{\beta}$.

**Example 1.** *Example 3.1 (Empirical RD_{RH} and PRD). What do empirical $RD_{RH}$ and PRD look like? To answer the question, 30 random bivariate standard normal points are generated (plotted in Figure 1) and $RD_{RH}$ and PRD are computed w.r.t. these points.*

*We select 961 equally spaced grid points from the square of $[x, y]$ with range of $|x| \le 3$ and $|y| \le 3$, then treat each point $(x, y)$ as a $\boldsymbol{\beta}^\top = (\beta_1, \beta_2)$ and compute its regression depth ($RD_{RH}$ and PRD) w.r.t. the 30 bivariate normal points. The depths of these 961 points are plotted in Figure 2.*

Inspecting the Figure reveals that (i) sample $RD_{RH}$ function is a step-wise increasing function (each step in this case is $1/30$). For this roughly symmetric data case, it can attain maximum depth around the center of symmetry (the origin), while (ii) on the other hand, PRD is a strictly monotonically increasing function and attains its maximum value at the center of symmetry, sharply contrasting the behavior of $RD_{RH}$ around the center (one has a unique maximum depth point and the other is opposite (multiple maximum depth points)).

## scatterplot of 30 normal points

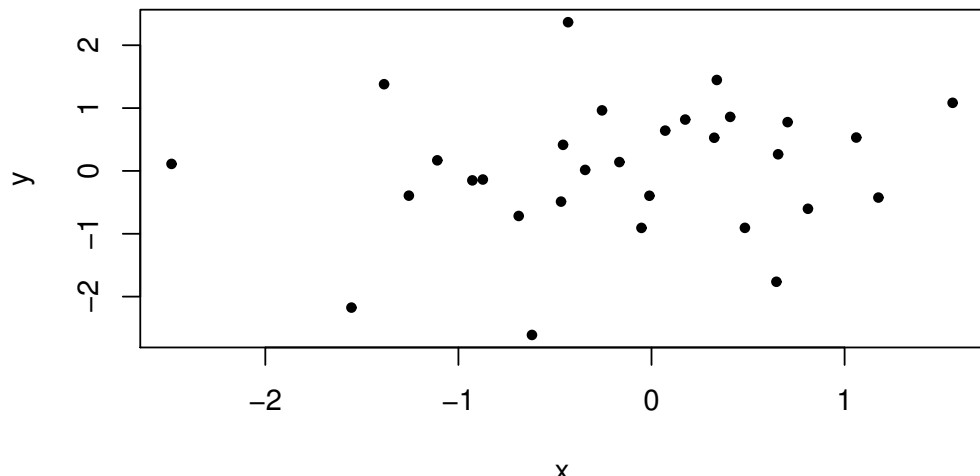

**Figure 1.** Thirty bivariate standard normal points.

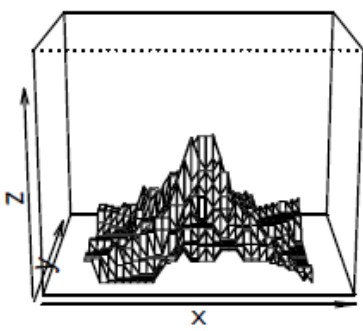
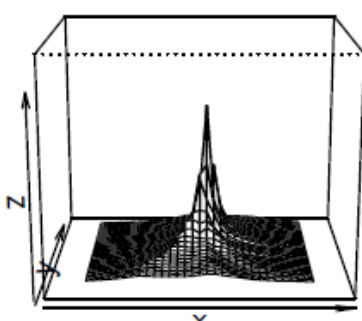

**Figure 2.** Regression depth (RD$_{RH}$) (**left**) and projection regression depth (PRD) (**right**) of 961 candidate parameter $\beta^{\top}$'s w.r.t. 30 bivariate standard normal points.

**Example 2.** ***Uniqueness of medians induced from empirical RD$_{RH}$ and PRD*** *This example illustrates the uniqueness behavior of the regression depth (RD$_{RH}$ and PRD)-induced medians in the empirical distribution case via a concrete example on the real data from the Hertzsprung–Russell diagram of the star cluster CYG OB1 (see Table 3 in chapter 2 of [16]), which contains 47 stars in the direction of Cygnus. Here, x is the logarithm of the effective temperature at the surface of the star ($T_e$), and y is the logarithm of its light intensity ($L/L_0$); see Figure 3 for the plot of the data set.*

Five regression lines are plotted in Figure 3. Among them, three (dashed red, dotted blue, and dotdash green) are regression medians from RD$_{RH}$, one (solid black) from PRD, and the other (longdash purple) is the least squares line. Note that the classical least squares regression estimator (as well as many traditional regression estimators) could be regarded as a depth-induced median under the general "objective depth" $D_{Obj}$ framework (see [11]). Thus, for the benchmark purpose, the least squares line is also plotted in Figure 3 alongside the four other median lines.

The LS line also justifies the legitimacy of the existence of RD$_{RH}$- and PRD-induced medians (as robust alternatives) since the LS line fails to capture the main-sequence/pattern of the data cloud (stars) and is heavily affected by four giant stars whereas the other four depth medians resist the four leverage points (outliers) and catch the main trend/cluster.

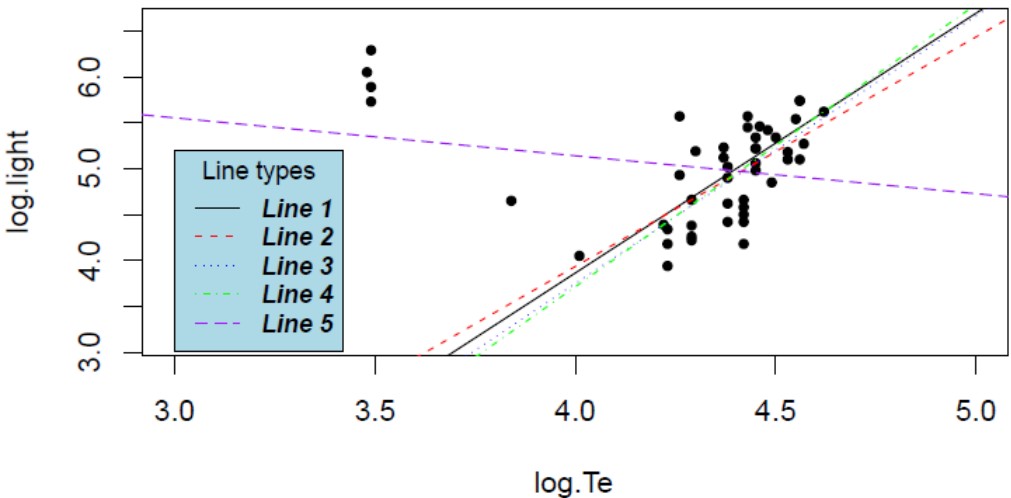

**Figure 3.** Five regression depth median lines based on the data from Hertzsprung–Russell diagram of the star cluster CYG OB1 (solid black for $\beta^*_{PRD}$; dashed red, dotted blue, and dotdash green all for $\beta^*_{RD}$; longdash purple for LS).

It turns out that there exist three maximum depth lines (medians) induced from $RD_{RH}$. Each of the three lines goes through exactly two data points. In terms of (intercept, slope) form, they are $(-6.065000, 2.500000)$, $(-8.586500, 3.075000)$, and $(-7.903043, 2.913043)$. These lines are plotted by dash red, dotted blue, and dotdash green in Figure 3. All three possess regression depth 21/47. Note that the average of the three deepest lines is $(-7.518181, 2.829348)$, which possesses RD 20/47. *That is, it no longer possesses the maximum regression depth.*

On the other hand, there exists only one maximum regression line (median), $(-7.453665, 2.829416)$, induced from PRD, plotted in solid black in Figure 3, with PRD value 0.8585901. Incidentally, the LS line is $(6.7934673, -0.4133039)$, plotted in longdash purple.

The computation issues of $RD_{RH}$ have been discussed in RH99, Rousseeuw and Struyf (1998) [20], and Liu and Zuo (2014) [21]. For the discussion on the computation of the PRD and induced regression medians, see Zuo (2019b) (Z19b) [22].

After obtaining $(\widehat{\beta}_1, \widehat{\beta}_2)$, one can immediately get the fitted line $\hat{y} = \widehat{\beta}_1 + \widehat{\beta}_2 x$ (which has actually already been plotted in Figure 3), and the predicted values: $\hat{y}_i = \widehat{\beta}_1 + \widehat{\beta}_2 x_i$, and hence the residuals: $r_i := y_i - \hat{y}_i$. All these involve the uniqueness issue, we first need to have a unique fitted line for each method. Here due to the non-uniqueness of the deepest RD lines, we select the first deepest line $(-6.065000, 2.500000)$ among the three as a representative. Then we construct a table with nine columns: 1st is the id's of observations, 2nd is the explanatory variable $x_i$ values, 3rd is dependent variable $y_i$ values, 4th–6th are the predicted $\hat{y}_i$ values for LS, RD, PRD methods, respectively, 7th–9th are the residuals $r_i$ for LS, RD, PRD methods, respectively (Table 1).

Next the residuals of three methods are plotted below in the Figure 4.

Table 1. Residuals analysis for three regression methods.

| i | x | y | $\hat{y}$ ls | rd | prd | r ls | rd | prd |
|---|---|---|---|---|---|---|---|---|
| 1 | 4.37 | 5.23 | 4.987329 | 4.860 | 4.910883 | 0.2426707 | 0.370 | 0.3191171 |
| 2 | 4.56 | 5.74 | 4.908802 | 5.335 | 5.448472 | 0.8311985 | 0.405 | 0.2915280 |
| 3 | 4.26 | 4.93 | 5.032793 | 4.585 | 4.599647 | −0.1027927 | 0.345 | 0.3303528 |
| 4 | 4.56 | 5.74 | 4.908802 | 5.335 | 5.448472 | 0.8311985 | 0.405 | 0.2915280 |
| 5 | 4.30 | 5.19 | 5.016261 | 4.685 | 4.712824 | 0.1737395 | 0.505 | 0.4771762 |
| 6 | 4.46 | 5.46 | 4.950132 | 5.085 | 5.165530 | 0.5098681 | 0.375 | 0.2944696 |
| 7 | 3.84 | 4.65 | 5.206380 | 3.535 | 3.411292 | −0.5563803 | 1.115 | 1.2387076 |
| 8 | 4.57 | 5.27 | 4.904668 | 5.360 | 5.476766 | 0.3653315 | −0.090 | −0.2067661 |
| 9 | 4.26 | 5.57 | 5.032793 | 4.585 | 4.599647 | 0.5372073 | 0.985 | 0.9703528 |
| 10 | 4.37 | 5.12 | 4.987329 | 4.860 | 4.910883 | 0.1326707 | 0.260 | 0.2091171 |
| ⋮ | ⋮ | ⋮ | ⋮ | ⋮ | ⋮ | ⋮ | ⋮ | ⋮ |
| 45 | 4.55 | 5.54 | 4.912935 | 5.310 | 5.420178 | 0.62706545 | 0.230 | 0.1198222 |
| 46 | 4.45 | 4.98 | 4.954265 | 5.060 | 5.137236 | 0.02573506 | −0.080 | −0.1572362 |
| 47 | 4.42 | 4.50 | 4.966664 | 4.985 | 5.052354 | -0.46666406 | −0.485 | −0.5523537 |

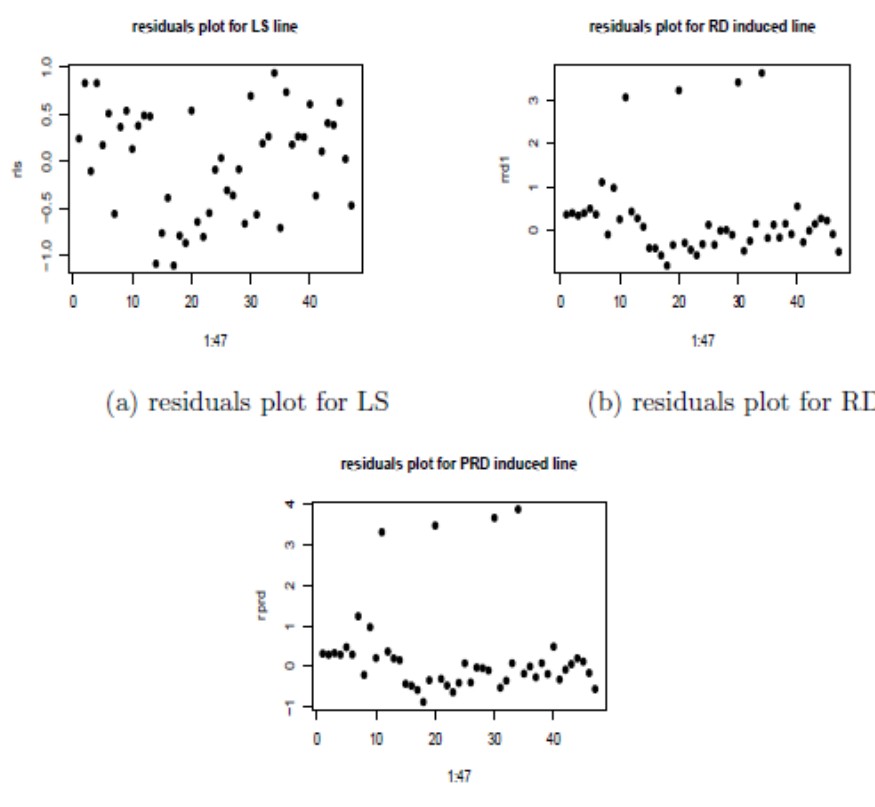

(a) residuals plot for LS                                                    (b) residuals plot for RD

(c) residuals plot for PRD

**Figure 4.** Residuals plots for three types of regression methods for the star cluster data. (**a**) Residuals plot for the LS method. (**b**) Residuals plot for one of deepest line induced from RD. (**c**) Residuals plot for the unique deepest line induced from PRD.

Inspecting the residuals plot immediately reveals that in this case, the residuals of LS method are rather deceptively homogeneous, its plot fails to identify any outliers whereas the robust regression median lines all can easily spot the four obvious outliers and two groups of stars. Based on the residual plot, one can make some conclusions. For example, the four outliers are not necessarily errors but might be exceptional observations (they come from a different group of stars), and LS line does not provide a good fit (only explained 4.4% of total variation in observations of *y*).

In the empirical distribution case, one can always take the average over all regression medians to take care of the non-uniqueness issue. Nevertheless, challenges arise computationally if there exist infinitely many medians in higher dimensions. Furthermore, the average sometimes will no longer be a deepest line/hyperplane (as seen in this example and more in Section 4).

The non-uniqueness issue is more vital with the population case since without the uniqueness, there will be no uniquely defined median and it is impossible to discuss the convergence (or consistency) and the limiting distribution of the unique empirical regression median.

## 4. Uniqueness of Regression Medians

From the empirical example in the previous section, we see that there can exist multiple empirical regression medians induced from $RD_{RH}$ while in the case of PRD there exists a unique one. These results are just empirical special examples and not for general cases. In the following, we address general cases and draw general conclusions.

### 4.1. Non-Uniqueness of $\beta^*_{RD_{RH}}$ and $\beta^*_{D_C}$

Under certain symmetry assumption (e.g., regression symmetry of Rousseeuw and Struyf (2004) (RS04)) [19] and other conditions, the regression median induced from $RD_{RH}$ can be unique (see Theorem 3 and Corollary 3 of [19]). However, generally speaking, we have

**Proposition 2.** $\beta^*_{RD_{RH}}(F_{(y, \mathbf{x})})$ *is not unique in general. The average of all* $\beta^*_{RD_{RH}}(F_{(y, \mathbf{x})})$ *might not possess the maximum depth any more.*

**Proof.** A counterexample suffices.

In fact, the real data example 3.2 could serve as one counterexample, where one has three maximum depth lines and the average line no longer possesses the maximum $RD_{RH}$ value.

An even simpler counterexample could be constructed. Assume that there are three sample points $A = (-1, 0)$, $B = (0, 1)$ and $C = (1, 0)$. Then it is readily seen that three lines each of which formed by two sample points are $(1, -1)$; $(1, 1)$ and $(0, 0)$ in terms of (intercept, slope) form and each line has the maximum $RD_{RH}$ $2/3$ whereas the average of all maximum depth lines is $(2/3, 0)$ which has $RD_{RH}$ only $1/3$. □

For special distributions, the median induced from Carrizosa depth can also be unique. But generally speaking, it is not.

**Proposition 3.** $\beta^*_{D_C}(F_{(y,\mathbf{x})})$ *is not unique in general. The average of all* $\beta^*_{D_C}(F_{(y,\mathbf{x})})$ *might not possess the maximum depth value any more.*

**Proof.** A counterexample suffices.

Denote by $H_\beta$ the hyperplane determined by $y = \mathbf{x}\top\beta$ for any $\beta \in \mathbb{R}^p$ and by $\theta_\beta$ the acute angle formed between the hyperplane $H_\beta$ and the horizontal hyperplane $H_h$ (y=0).

Assume that $\beta_i \in \mathbb{R}^p$, $(i = 1, 2)$, $\beta_1 \neq \beta_2$, and $H_{\beta_i}$ each contains $1/2$ probability mass; any hyperline in $H_{\beta_i}$ contains no probability mass ($i = 1$ or 2); $\theta_{\beta_1} = \theta_{\beta_2}$, and $H_{\beta_1}$ intersects with $H_{\beta_2}$ at a hyperline in the horizontal hyperplane $H_h$.

Now in light of characterization (5) of $D_C$, it is readily seen that at each $\beta_i$, $D_C(\beta_i; P)$ attains the maximum depth value $1/2$.

Let $\gamma = (\beta_1 + \beta_2)/2$, then it is readily seen that the $D_C(\gamma; P) = 0$, and $H_\gamma$ is no longer a hyperplane with the maximum depth value. □

*4.2. Uniqueness of $\beta^*_{PRD}$*

For two univariate random variables $X, Y$ defined on the sample space $\Omega$, $X < Y$ stands for $X(\omega) < Y(\omega)$, $\forall \omega \in \Omega$. We say that $T(F_{(y-\mathbf{x}^\top\beta)/\mathbf{x}^\top\mathbf{v}})$ is *strictly monotonic* at point $\beta_0$ iff $T(F_{(y-\mathbf{x}^\top\beta_0)/\mathbf{x}^\top\mathbf{v}}) > T(F_{(y-\mathbf{x}^\top\beta_1)/\mathbf{x}^\top\mathbf{v}})$ whenever $-\mathbf{x}^\top\beta_0 > -\mathbf{x}^\top\beta_1 \ \forall \beta_1 \in \mathbb{R}^p$, for any $\mathbf{v} \in \mathbb{S}^{p-1}$.

**Proposition 4.** *If (A0) holds and $T(F_{(y-\mathbf{x}^\top\beta)/\mathbf{x}^\top\mathbf{v}})$ (i) is strictly monotonic at $\mathbf{0}$ and (ii) satisfies (A1), (A2), and (A3), then $\boldsymbol{\beta}^*_{PRD}(F_{(y, \mathbf{x})})$ exists uniquely.*

**Proof.** To prove the proposition, we first invoke the following result. $\square$

**Lemma 1** ([11]). *The PRD and $\boldsymbol{\beta}^*_{PRD}$ satisfy the following propoerties.*

(i) *The $\boldsymbol{\beta}^*_{PRD}(F_{(y, \mathbf{x}^\top)})$ is regression, scale and affine equivariant in the sense that*

$$\boldsymbol{\beta}^*(F_{(y+\mathbf{x}^\top\mathbf{b}, \mathbf{x}^\top)}) = \boldsymbol{\beta}^*(F_{(y, \mathbf{x}^\top)}) + \mathbf{b}, \ \forall \mathbf{b} \in \mathbb{R}^p;$$

$$\boldsymbol{\beta}^*(F_{(sy, \mathbf{x}^\top)}) = s\boldsymbol{\beta}^*(F_{(y, \mathbf{x}^\top)}), \ \forall \ scalar \ s(\neq 0) \in \mathbb{R};$$

$$\boldsymbol{\beta}^*(F_{(y, A^\top\mathbf{x})}) = A^{-1}\boldsymbol{\beta}^*(F_{(y, \mathbf{x}^\top)}), \ \forall \ nonsingular \ A \in R^{p \times p},$$

*respectively.*

(ii) *The maximum of $PRD(\boldsymbol{\beta}; F_{(y,\mathbf{x}^\top)})$ exists and is attained at a $\boldsymbol{\beta}_0 \in \mathbb{R}^p$ with $\|\boldsymbol{\beta}_0\| < \infty$.*

(iii) *The $PRD(\boldsymbol{\beta}; F_{(y,\mathbf{x}^\top)})$ monotonically decreases along any ray stemming from a deepest point in the sense that for any $\boldsymbol{\beta} \in \mathbb{R}^p$ and $\lambda \in [0,1]$,*

$$PRD(\lambda\boldsymbol{\beta}^* + (1-\lambda)\boldsymbol{\beta}; F_{(y, \mathbf{x}^\top)}) \ \geq \ PRD(\boldsymbol{\beta}; F_{(y, \mathbf{x}^\top)}),$$

*where $\boldsymbol{\beta}^*$ is a maximum depth point of $PRD(\boldsymbol{\beta}; F_{(y,\mathbf{x}^\top)})$ for any $\boldsymbol{\beta} \in R^p$.*

Now we are in a position to prove the proposition.

Assume, w.l.o.g., that $S(F_y) = 1$ (since it does not involve $\mathbf{v}$ and $\boldsymbol{\beta}$, it has nothing to do with the maximum depth point $\boldsymbol{\beta}^*_{PRD}$). The existance of the maximum depth point (the regression median) is guaranteed in light of Lemma 4.1 above. We thus focus on the uniqueness. Assume that there are two maximum depth points $\boldsymbol{\beta}^*_1 \neq \boldsymbol{\beta}^*_2$. We seek a contradiction.

Let $\boldsymbol{\beta}^*_0 = (\boldsymbol{\beta}^*_1 + \boldsymbol{\beta}^*_2)/2$. By virtue of Lemma 4.1 above, $\boldsymbol{\beta}^*_0$ is also a maximum depth point. By the invariance of the projection regression depth functional (see [11]) and Lemma 4.1 above, assume (w.l.o.g.) that $\boldsymbol{\beta}^*_0 = \mathbf{0}$.

For a given $\boldsymbol{\beta} \in \mathbb{R}^p$, write $g(\boldsymbol{\beta}, \mathbf{v}) := T(F_{(y-\mathbf{x}^\top\boldsymbol{\beta})/\mathbf{x}^\top\mathbf{v}})$. In light of the continuity of $T$ in $\mathbf{v}$, the generalized extreme value theorem on a compact set, and **(A1)**, there exists a $\mathbf{v}_\beta \in \mathbb{S}^{p-1}$ such that

$$g(\boldsymbol{\beta}, \mathbf{v}_\beta) = \sup_{\mathbf{v} \in \mathbb{S}^{p-1}} |T(F_{(y-\mathbf{x}^\top\boldsymbol{\beta})/\mathbf{x}^\top\mathbf{v}})| \tag{14}$$

For simplicity, denote by $\mathbf{v}_0$ for $\mathbf{v}_{\beta^*_0}$. Then we have

$$g(\boldsymbol{\beta}^*_0, \mathbf{v}_0) = T(F_{(y-\mathbf{x}^\top\boldsymbol{\beta}^*_0)/\mathbf{x}^\top\mathbf{v}_0}) = T(F_{y/\mathbf{x}^\top\mathbf{v}_0}) = \sup_{\mathbf{v} \in \mathbb{S}^{p-1}} |T(F_{(y-\mathbf{x}^\top\boldsymbol{\beta}^*_0)/\mathbf{x}^\top\mathbf{v}})| := \alpha^*. \tag{15}$$

Denote by $l(\boldsymbol{\beta}^*_1, \boldsymbol{\beta}^*_2)$ the hyperline that connects $\boldsymbol{\beta}^*_1$ and $\boldsymbol{\beta}^*_2$ in the parameter space of $\boldsymbol{\beta} \in \mathbb{R}^p$. Consider two cases.

**Case I x** *does not concentrate on any single hyperplane.* In light of this assumption, there exists at least one $\gamma \in \mathbb{R}^p$ on $l(\boldsymbol{\beta}_1^*, \boldsymbol{\beta}_2^*)$ in the parameter space $\mathbb{R}^p$ such that $-\mathbf{x}\top\boldsymbol{\gamma} \neq 0$. Assume (w.l.o.g.) that $-\mathbf{x}\top\boldsymbol{\gamma} < 0 = -\mathbf{x}\top\boldsymbol{\beta}_0^*$. By (15) and the strictly monotonicity of $T$, one has that for the $\mathbf{v}_\gamma$ defined in (14)

$$\alpha^* = \inf_{\boldsymbol{\beta} \in \mathbb{R}^p} \sup_{\mathbf{v} \in \mathbb{S}^{p-1}} |T(F_{(y-\mathbf{x}\top\boldsymbol{\beta})/\mathbf{x}\top\mathbf{v}})| \leq \sup_{\mathbf{v} \in \mathbb{S}^{p-1}} |T(F_{(y-\mathbf{x}\top\boldsymbol{\gamma})/\mathbf{x}\top\mathbf{v}})| = T(F_{(y-\mathbf{x}\top\boldsymbol{\gamma})/\mathbf{x}\top\mathbf{v}_\gamma})$$

$$< T(F_{(y-\mathbf{x}\top\boldsymbol{\beta}_0^*)/\mathbf{x}\top\mathbf{v}_\gamma}) \leq \sup_{\mathbf{v} \in \mathbb{S}^{p-1}} |T(F_{y/\mathbf{x}\top\mathbf{v}})|$$

$$= T(F_{y/\mathbf{x}\top\mathbf{v}_0}) = \alpha^* \tag{16}$$

which is a contradiction. This completes the proof of the Case I.

**Case II x** *concentrates on a single hyperplane.* This implies that there is a $\mathbf{v} \in \mathbb{S}^{p-1}$ such that $\mathbf{x}\top(\omega)\mathbf{v} = 0$ for any $\omega \in \Omega$. This contradicts **(A0)**, however. This completes the proof of Case II and thus the proposition.

**Remark 1.** *(I) **(A0)** automatically holds if **x** has density or if **x** is not degenerated to concentrate on a single $(p-1)$ dimensional hyperplane. The latter means all x lie on the same point for $p = 1$, and they lie on a single line for $p = 2$, and lie on a plane for $p = 3$, and so on.*
*(II) **(A1)**, **(A2)** and **(A3)** hold for a large class of T, such as the mean, weighted mean (Wu and Zuo (2009) [23]), and quantile functionals.*
*(III) There also exists a large class of T that is strictly monotonic. For example (i) If $T(F_{(y-\mathbf{x}\top\boldsymbol{\beta})/\mathbf{x}\top\mathbf{v}}) = E\left((y - \mathbf{x}\top\boldsymbol{\beta})/\mathbf{x}\top\mathbf{v}\right)$, then T is strictly monotonic at any $\boldsymbol{\beta}$ as long as the related expectations exist and $E(\mathbf{x}\top\boldsymbol{\alpha}/\mathbf{x}\top\mathbf{v}) > 0$ whenever $\mathbf{x}\top\boldsymbol{\alpha} > 0$ for any $\boldsymbol{\alpha} \in \mathbb{R}^p$ and $\mathbf{v} \in \mathbb{S}^{p-1}$. (ii) When $T(F_{(y-\mathbf{x}\top\boldsymbol{\beta})/\mathbf{x}\top\mathbf{v}}) = Q_q\left((y - \mathbf{x}\top\boldsymbol{\beta})/\mathbf{x}\top\mathbf{v}\right)$, $q \in (0,1)$, where $Q_q(Z)$ is the qth quantile associated with the random variable Z (i.e., $Q_q(Z) = \inf\{z : P(Z \leq z) \geq q\}$), then T is strictly monotonic at any $\boldsymbol{\beta}$ as long as the CDF of $Z(\boldsymbol{\beta}; \mathbf{v}, y, \mathbf{x}) := (y - \mathbf{x}\top\boldsymbol{\beta})/\mathbf{x}\top\mathbf{v}$ is not flat at $\boldsymbol{\beta}$ for a given $\mathbf{v} \in \mathbb{S}^{p-1}$.*
*(IV) The proposition covers the sample case. That is, when $F_{(y,\mathbf{x}\top)}$ is replaced by its sample version in the proposition, we have the uniqueness of the sample regression median induced from PRD, which is very helpful in the practical computation of the median and consistent with the finding in Figure 2.*

## 5. Concluding Remarks

### 5.1. Why Do We Care About the Non-Uniqueness of Regression Medians?

Uniqueness is actually implicitly assumed when we discuss the property (such as the Fisher consistency, regression, scale and affine equivariance, or asymptotic breakdown point) of regression medians. Without the uniqueness, (i) the sample regression median can never converge in probability or in distribution to its population version, (ii) deepest regression will yield more than one response and residual for a given **x**, (iii) algorithms for the approximate computation of sample medians can never converge.

Uniqueness is so essential in our discussion of medians that there is a conventional remedy measure for non-uniqueness: to take average of all medians. This works in many scenarios, but not for $\boldsymbol{\beta}_{RD_{RH}}^*$ and $\boldsymbol{\beta}_{D_C}^*$. This phenomenon for $\boldsymbol{\beta}_{RD_{RH}}^*$ was first noticed by Mizera and Volauf (2002) [24] and Van Aelst et al. (2002) [25]. Concrete examples such as the real data Example 3.2 and the artificially constructed one in the proof of Proposition 4.1 are presented here though.

### 5.2. Why Do We Just Treat Three Regression Medians?

$D_C$ ([12]) and $RD_{RH}$ ([10]) are two pioneer notions of regression depth. PRD was recently introduced in [11]. The latter systematically studied the three regression depth notions w.r.t. four axiomatic properties, that is, **(P1)**, **(P2)**, **(P3)** and **(P4)** (see Section 2). It is found out that both regression

depth $RD_{RH}$ and projection regression depth PRD are real depth notions in regression since both satisfy **(P1)–(P4)**. While the former needs an extra assumption **(A)** (see Section 2.2), the latter does not need any extra assumptions. On the other hand, Carrizosa depth $D_C$ violates **(P3)** in general, hence is not a real regression depth notion w.r.t. the definition in [23]. That motivates us to just focus on $RD_{RH}$ and PRD throughout.

*5.3. Summary and Conclusions*

In terms of robustness, both depth-induced medians are indeed robust. In fact, the median $\beta^*_{RD_{RH}}$ can asymptotically resist up to 33% [13] contamination, whereas $\beta^*_{PRD}$ can resist up to 50% [14] contamination without breakdown, sharply contrasting to the 0% of the classical LS estimator.

In terms of efficiency, sample $\beta^*_{PRD}$ could possess a higher relative efficiency when compared with sample $\beta^*_{RD_{RH}}$ (see [22]).

Now in terms of uniqueness, $\beta^*_{PRD}$ again distinguishes itself from the leading depth median $\beta^*_{RD_{RH}}$ by generally possessing the desirable uniqueness property.

From the computational point of view, RD (and $\beta^*_{RD_{RH}}$) has an edge over PRD (and $\beta^*_{PRD}$). The former is relatively easier to compute than the latter (see [22]).

By virtue of the performance criteria above, we conclude that PRD and $\beta^*_{PRD}$ are promising options among the leading regression depths and their induced medians.

**Funding:** This research received no external funding.

**Acknowledgments:** The author thanks Hanshi Zuo for his careful proofreading of the manuscript and two anonymous referees for their helpful and constructive comments and suggestions, all of which have led to improvements in the manuscript. Special thanks go to the Managing Editor, Yuanyuan Yang for her enthusiastic invitation, encourage, and full support.

**Conflicts of Interest:** The authors declare no conflict of interest.

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
