# Peer review of "Depth Induced Regression Medians and Uniqueness"

_stats, doi:10.3390/stats3020009_

Round 1

Reviewer 1 Report

The paper is well written and organized, I found only "technical" problems 1) I did not find abstract, title and keywords, 2) I could not see Figures 2 and 3.

Minor comments:

line 13: It might be usefull to recall how the quantile is defined. What if we consider a uniform distribution on (0,1)U(2,3). What is the value of median in this case?

line 16: brackets: (Donoho (1982)) - delete two additional )(

line 21: and -> an

line 64-65: monotonicity relative to the deepest point,

line 85: bracket ) in lower index is missing

line 106: for thought: What can be said about theoretical (not sample) medians?

line 122: How instead of What ?

line 156: comma missing in (-7.4, 2.8)

line 170: previous instead of last ?

line 170: multiple

line 211: a bit clumsy expression "bounded beta_0"

line 213-214: What is G? There should be PRD on both sides.

line 214: PRD(beta*,...) ?

line 214: Why the symbol of square?

lines 218, 220, 221: Lemma 4.1, not 3.1

line 219: contradiction

line 232: (A2)

Author Response

Please see the attached file for my point-by-point response to your report.

For your convenience, the revised manuscript has also merged with the response

file.

Reviewer 2 Report

"Regression medians and uniqueness" by Zuo (2020)

The author presented some interesting uniqueness properties of Depth Regression medians modelling. Depth feature serves as a good alternative for classical least squares estimator. The author presented in clear and well form the propositions. However, the examples lack of statistical analysis: fit, estimation, residuals inspection, etc.

I recommends to author to improve this aspect to complete the manuscript, and submit to STATS again. Then I could review a new version of this manuscript.

Some minor comments:

  1. Improve the title, please do it more representative. For example: "Depth induced Regression medians and uniqueness".
  2. P1 L8: DC not defined.
  3. Eq (1): write the regression with the observations i=1,...,n.
  4. P2 L38: Use \top for transposed vector/matrix.
  5. Fix the reference's citations in the body's manuscript.

Author Response

Please see the attached my point-by-point responses to your report.

For your convenience, the revised manuscript has also provided at the end of the response file. 

Round 2

Reviewer 2 Report

2nd review of: "Depth induced regression medians and uniqueness" by Zuo (2020)

In this new review, I see that author addressed several aspects related to my last review. However, I don't convinced with their reply to my first question. He said: "... these tedious details do not deserve to be published and thus are
omitted..."

I don't believe that these "details" are tedious or can be ommited. In any new modelling, it is important to shows the model's development on a real-life dataset. For this reason, I cannot recommends the publication of this manuscript is STATS.  

Author Response

please see the attached file of my response to the reviewer

Round 3

Reviewer 2 Report

Dear author,

thanks for provides final details about regression. Now, I could recommend the publication of this manuscript. I have only 1 minor comment:

- Figure 3, in title: fix "medains" by "medians". Also, put ix X and Y axis: log(Te) and log(Light).